# Numerical Case Studies about Two-Dimensional CHS Joints with Symmetrical Full-Overlapped Top-Connection

**DOI:** 10.3390/ma15093333

**Published:** 2022-05-06

**Authors:** Patrick Heinemann, Dorina-Nicolina Isopescu

**Affiliations:** Department of Civil and Industrial Engineering, Faculty of Civil Engineering and Building Services, “Gheorghe Asachi” Technical University of Iasi, 1, Prof. Dr. Docent Dimitrie Mangeron Blvd., No. 59A, 700050 Iasi, Romania

**Keywords:** hollow sections, FEM analysis, welding line, construction, steel

## Abstract

Steel joints made out of circular hollow section profiles are used for many fields of applications, such as wide-span or representative halls for airports. By connecting two inclined pipes to a vertical third pipe, a ramified vertical column is created, where the node is the weakest point in the structure due to the geometrical heterogeneity. The current standards and Design Codes have limitations regarding the geometrical properties of hollow sections joints. However, the kind of steel joint presented in this paper is excluded in the current standards. This paper is about numerical FEA case studies of two-dimensional, circular hollow-section joints to figure out the resistance of atypical steel joints. In the first step, a small-scale model is generated to analyze the influence of the inclination. In the second step, the geometries of the different pipes are extended. The influence of the inclination angle and the stability of the joint are analyzed. It was discovered that the inclination angle between the three pipes has a large influence on the stresses and deflections at the node. By increasing the inclination angle, the maximum applied force can be increased. The extended members change the behavior and the stress distribution.

## 1. Introduction

Structures made of hollow-section steel profiles are mainly used for two reasons. Firstly, there are aesthetical and architectural aspects. Secondly, there are physical advantages, such as high rotational stiffness and low weight. In widespan structures and large representative halls, such as airports or universities, designing columns are installed to support the roof structure. Some of these columns are built as a combination of two or more single members. A commonly used design is the full-overlapped column with on-top connection. A steel plate is connected to a vertical pipe. Two or more inclined pipes are connected onto this plate. A “Y-shape” is generated, which differs to the Y-shape in current standards. There are uniplanar (two-dimensional) or multiplanar (three-dimensional) versions. The connection between the steel members is either done by a three-dimensional welding line or cast steel. This paper is about the welding line version. The node is defined as a position where the lines of application strike together. Designing the nodes of the column, including the welding line is a complex process. Due to the geometrical and material heterogeneity, the node is the weakest part of the column. Current standards and design codes, such as the Eurocode 3 [1] or CIDECT [2], have limitations regarding the material, load cases, or geometry. Steel joints with inclination angles (angle between the upper pipe and the horizontal level) of less than 30° are offset from the standard. Current Design Codes adopt joints where one or more pipes are connected to the side flank of a third pipe. Joints with a connection at the end of a pipe are not covered in the standards. Designing engineers have to create their own models or execute experimental tests for each geometrical variation to design columns with full-overlapped geometries. The aim of a designing engineer is to choose on one hand the highest resistance variant and on the other hand the most economical variant. By varying the inclination angle, the resistance of the joint is changed and the structure can be optimized economically. The overall column length is determined by the height of the hall. The ratio of the length between the members is chosen by the designing engineer. The case studies in this paper analyze the effect of varying the pipe length to archive an economical design of the structure. The novelty of this study is about a commonly used special steel joint, which is undefined in current standards. This article helps the designing engineer to choose the most economical structure under the aspects of chord length ratio and inclination angle. 

It is hard to find comparable studies in the literature, which shows the relevance of the analysis. Mainly, the analyzed steel joints have a standard defined geometry. Parallels in the process of analyzation, similar steel joints, or numerical analyzations in the field of the full-overlapped Y-joint are presented. 

Variations of joints including similar geometrical characteristics were analyzed by Heinemann et al. in a multiplanar [3] or uniplanar way [4,5,6]. It was found that there are large differences in the resistance of the nodes. The highest stresses arose at the welding line. Azari-Dodaran et al. [7] extended the observation and explained that there are significant differences in the resistance between uniplanar and multiplanar steel joints for different load scenarios under extraordinary conditions. He performed investigations of steel joints under high temperatures. Jankovic [8] conducted studies about brace to chord connections concerning a Y-joint. A calculation method was shown where the influence of a second pipe on a first pipe was calculated. A substitute force affected the pipe. The focus was on the designing of the pipe. The inclined force was transferred into a local stress at the side of the first pipe. Mia et al. [9] performed a numerical analysis about special steel joints, defined as XT-joints. There was no overlapping node, but a single pipe was affected by more than two other pipes. The inclination angle was given as 90°. There was no variation of the inclination angle. The cross-sections and mesh dimensions are comparable to the model in this paper. A focus was given to the pipe thickness. The load case was a combination of compressional, tensional, and bending forces. It was found that the stress and deflection distribution at the connection of the pipes is not linear. No fracture arose, and the analysis was done in the elastic state. Rezadoost et al. [10] conducted studies about X-joints, where the joint area is reinforced. The reinforcement adopted the shape of the joint and was made of welded tubes. Especially the effect and behavior of the reinforced part is comparable to the analyzed joint in this paper. The X-joint has an inclination angle of 90°. A fillet weld was set to the model. The mesh is a combination of shell and solid elements. A focus is on the variation of geometrical parameters, e.g., the diameter ratio of the pipes. As it is done for the full-overlapped study in this paper, there are pipes with smaller and larger cross-sections. It was found that there is a large influence of the diameter ratio on the resistance—in this case, on the fatigue resistance. Advanced numerical analyses are shown to underline the importance of finite element studies regarding civil engineering structures. Kolanu et al. [11] did finite element studies on welded T-panels under compression. It was figured out that the computational time increases exponentially if the mesh size decreases. A difference of 4% between the numerical and experimental deformations occurred. Horajski et al. [12] carried out advanced FEM studies on welded thin-walled structures. A numerical algorithm was found to adopt the welding process. Vakili-Tahami et al. [13] performed FEA studies on welded T-joints with steel panels. Several plate thicknesses were analyzed. It was found that the differences between the numerical and experimental results depend on the plate thicknesses and welding direction. 

This paper is about numerical case studies of uniplanar joints made of circular hollow section profiles (CHS) with full-overlapped top connection. The aim is to figure out the influence of the inclination angle on the stability of the column. With the results, the designing engineer will have guidance in their decision to choose an economic inclination angle without executing a laboratory test. The paper is split into two parts. Firstly, there is a case study of a small-scale model. The pipe length was shortened to neglect the influences of stability failure. The focus is on the failure of the welding line instead of the pipe member. The second part is about geometrically expanded models with different inclination angles. On one hand, the influence of the chord length on the maximum affected force on the joint is focused; on the other hand, the influence of the inclination angles on the maximum affected force is analyzed. Espinosa [14] showed that the influence of the chord length becomes negligible for a length-to-diameter ratio of larger than 3. This conclusion is analyzed for the special joint, defined as a full-overlapped Y-joint in Section 3.2 and Section 3.3. The ratio is given with 12 to 23, depending on the chord length. 

## 2. FEA—Model and Analysis 

The vertical pipe is defined as a chord and the two connected members are defined as braces or branches. The chord had an overall length of 350 mm and the braces were set to 175 mm, independent from the inclination angle. The vertical chord had a cross-section of 127.0 × 3.6 mm (DN 100), as defined in the EN 10219-2 [15]. At the chord’s top, there was a flat, 5-mm steel plate as an adapter between the braces and the chord. The two braces had a smaller diameter, including a cross-section of 51.0 × 2.6 mm (DN 40). This steel profile is defined in the standard EN 10219-2 [15]. In steel construction, different kinds of welding lines exist. Zamzami [16] explained the different parameters and influences of the welding line types; these are butt-, fillet-, or cruciform weldings. Saini et al. [17] explained that in numerical studies, the welding line does not to be modeled if there is no high bending. 

However, for this study, the welding line was set as a 3-mm fillet weld, which followed the shape of the braces and the top plate of the chord. Due to the complex shape, other welding line types are not realizable. In the view of assembling on the construction site, the members are typically arranged including a gap of 1–2 mm. By implementing the welding line, the pipes are connected via the welding line. So, there is no direct contact between the pipes and the force can only be transferred through the welding line. This presumes a precise preparation of the welding line, without deflects or imperfections. Comparable results between the numerical and experimental model were archived by robotic welding processing. By idealizing this to the numerical model, the contact between the braces and brace-to-chord was set as frictionless. The connection between the brace and weld or top plate to weld was set as fixed. The force was only transferred through the welding line. However, this is a numerical adoption of a realistic welding process. A connection between the pipes and the steel plate can only arise for lager deformations in the plastic state. The in-plane models have the same inclination angles of 30°, 45°, and 60° for both braces. Figure 1 visualizes the general model, including the design of the welding line. Beside this, the load case is shown. The axial compression force is visualized in this figure as an example of one inclination angle. 

The braces’ ends are simply supported. So, the displacement is blocked, while rotation is allowed. The displacement at the end of the chord, which is affected by the force, is blocked for the horizontal components. The vertical direction is unblocked. Typically, loads of a roof are constant and static loads, such as dead or snow loads; these loads are transferred as compression loads to the columns. The axial compression force is applied to the cross-section area of the chord in the dimension (N/mm^2^) to obtain a uniformly compression. The matched force in the results (Section 3) is the total force in the dimension (kN). The analysis is split into three parts. First, there is a case study of a small-scale model. In Section 3.2 and Section 3.3, there are case studies including expanded geometries. Regarding the small-scale model, a compression force affects the chord’s end in an axial direction. The compression load is iterated to the limit state (yield stress) of the material by increasing the force. The aim is to observe the influence of the inclination angle on the stresses and deflection in the welding line area. Cracks are the result of various load scenarios. They arise by exceeding the ultimate strength or durableness. If there are dynamic loads, fatigue cracks will appear. These multiaxial stresses in the area of the cracks are complex to calculate [18,19,20,21]. Even due to the geometrical heterogeneity of the node, a multiaxial state of stress arises [22]. 

The material was chosen as steel type S 235. The main properties of the material are as follows: yield stress is 250 MPa, Poisson’s ratio is 0.3, density is 78.5 kN/m^3^. The elastic model was analyzed. Heinemann et al. [23] and Younise et al. [24] analyzed the influence of materials regarding standard-defined nodes. Carbon, wood, aluminum, and different steel alloys were the subjects of the simulation. The differences in the results for the various steel alloys are small. An exception was found for the Aluminum material.

The software ANSYS [25] was used for the numerical simulation. The mesh was made of 10-mm tetrahedral solid elements with a quadratic element shape function. At the area of the welding line, there was a refinement of 1.3-mm elements. The refinement of the mesh ensures precise results at decisive positions, while the computational time is minimized. A focus was given to the welding line, including its three-dimensional shape. To obtain the best-fitting numerical adoption, solid elements instead of shell elements were chosen. Hobbacher [26] suggested an element size of “0.4 x t” for solid elements, regarding the design of a welding line. Heinemann et al. [5] conducted prestudies to figure out the influence of different mesh types on the stress distribution and deflection in case of Y-joints. It was found that the chosen mesh is precise and saves computational time. In Figure 3, the 5-mm top plate and the triangle shape of the fillet weld are shown. 

A mesh metric analyzation was performed on the models to evaluate the quality of the mesh. As an example, the evaluation of the 45° small-scale model is presented. The skewness tended to zero, and there was a value of 0.25 on average. The maximum value was smaller than 1. Problems regarding the aspect ratio occur for values larger than 1000. The maximum value for the aspect ratio in this case study was given as 104. The average value was 1.92. The Jacobian value was calculated by 5.4. The average value was 1.01. The common limit value was given as 30. The overall mesh quality was evaluated in the range of 0 to 1. A value of 0.819 was given for the quality on average. 

For the verification of the stress results, the following study was consulted. Heinemann et al. [27] carried out experimental and numerical studies about full-overlapped CHS steel joints without intermediate plates. The geometrical properties are similar to the small-scale model in Section 3.1. The numerical analyzation process is comparable and the mesh choice superimposable. The experimental tests were done for the verification of the numerical model by comparing the von-Mises stresses. Several strain gauges were fixed at the specimen. As the geometry and the load scenario were symmetrical, two strain gauges were set in-between both branches nearby the welding line. So, the position was at the top area of the vertical welding line. Strain gauge 2 was located on brace 1 and strain gauge 3 was located on brace 2. Due to the symmetry, both sensors recorded nearly the same strains. The strains are compared with the numerical and experimental analysis at the same position. As an example, concerning one position, Figure 2 shows the differences between the numerical and experimental results as an example for one strain gauge position (Pos. 2 and 3). The range of the inclination angles is enlarged. It was found that the numerical analysis is in good agreement with the experimental results. Especially, in the case of the 25° model, the numerical and experimental results are nearly superimposable. The range of differences is small. This validation study is seen as a pattern for the numerical analysis in this paper.

## 3. Results

### 3.1. FEM Analysis of the Small-Scale Joint

The first part of Section 3 is about small-scale CHS joints with full-overlapped top connection. The brace-to-chord ratio is fixed and given as 1:2. In Figure 3, the geometry and the mesh are visualized including the triangular shape of the welding line, which transfers the force. The welding line is connected to the 5-mm top plate. Next to the geometry, the von-Mises stress distribution is visualized on the surface of the model. The maximum stresses arise at the welding line (yellow or red parts) or at the contact area between the brace and the welding line.

The deflections result of the 30°, 45°, and 60° models have a similar distribution. Figure 4 shows the deformation of the CHS steel joint. The overall deformation is small by setting a true-to-scale representation. For the analyzation of the failure mode, an optical magnification factor of 280 is implemented to the graphic. The maximum deflections arise at the chord’s end. Due to the applied force, the largest axial compression deformation occurs at this point. The minimal deflections are at the welding line between both branches and on the top of the steel plate. Figure 5 shows the results of the maximum compression force, which is applied to the model to reach the elastic limit state of the joint. The graph has an increasing trend. The differences in the maximum compression force are small for 30° and 45° with values of 10.3 kN and 11.25 kN. By increasing the inclination angle to 60°, an axial compression force of 18.25 kN can be induced. However, by the implementation of a larger angle, a lower load is absorbed. There is a correlation between a higher applied compression force and the resistance of the joint. If the inclination angle is steep, the horizontal steel plate tends to deform largely, which results in higher stresses at the welding line.

In Figure 6, the deflection of the symmetrical CHS system is presented. The shape of the graph increases nearly linearly. The 30° model has the smallest deflections, while the 60° angle model has the highest. By increasing the inclination angle, the maximum applied force is increased by 85% and the deflection is increased by 27%.

### 3.2. FEM Analysis of the Expanded Geometry (Variable Geometry with Fixed Braces Length)

#### 3.2.1. Expanded 30° Model

To analyze the failure mode according to the stability, the length of the chord is expanded, while the braces’ lengths are fixed. Examples for failure by stability are the arising of buckling or bending effects at the pipes before reaching the ultimate limit state of the welding line. Generally, the length of the chord varies in the range of 1.5 m to 3.0 m for three different inclination angles. As an example of the general model, Figure 7 shows the 45° case. Figure 7a–h show the models including the variable chord length. The delta steps are 0.20 m with an exception for the first case, which is equal to 0.30 m (Figure 7b). To see the difference regarding the chord length, an equal scale is used in Figure 7. The vertical repositioning of the braces would change the constructive detail. By doing this, a K-joint is created, which is already defined in the Design Code Eurocode [1]. The load case is changed to the model in Section 3.1. 

Figure 8 shows the results for the maximum compression force. There is a decreasing trend, except in the 2.2 m and 3.0 m models. A similar shape is shown in Figure 9 regarding the results of the maximum deflections. The 1.5-m chord system has the highest deflection. Overall, the range of the deflections between the models is small. Generally, there is a maximum range of reduction regarding the compression force of −14% and −12% for the deflections in case of the 30° model.

The boundary conditions for the chord’s end and braces’ ends are set as simply supported. The braces are affected by axial compression forces, which are equal to both braces. Djokovic [28] explained the advantage of including bending moments in the load scenario. In future studies, bending load cases will be added. Case studies of expanded brace-lengths models are presented in Section 3.3. The maximum stresses arise at the welding line between both branches and at the outer edges of the welding. The results are presented in the elastic limit state, when the yield stress of the material is reached. There is no failure by stability of the pipes before the limit state of the welded node is reached. The resistance of the welding line is decisive. 

#### 3.2.2. Expanded 45° Model

This paragraph is about an expanded CHS joint in the version of the 45° inclination angle. The process of analyzation, load scenario, meshing, and profile dimensions is analogue to the 30° model in Section 3.2.1. Figure 10 shows the distribution of the von-Mises stresses in case of the 45° model. The maximum stresses arise at the side flank of the welding line, which is visualized in the yellow and red area. No failure by stability occurs due to the combination of the symmetrical geometry and load. The results for the maximum compression force and deflection are shown in Figure 11 and Figure 12. The graph of the compression force decreases nearly linearly. In contrast to this behavior, the results of the maximum deflections quadratically decrease. The system that includes the 1.5-m chord model has the highest deflections and stresses. The maximum compression force is reduced by a factor of 25% and 30% regarding the deflections, in case of the 45° inclination angle system.

#### 3.2.3. Expanded 60° Model

The third variation of an expanded CHS joint model with a fixed chord length includes the inclination angle of 60°. The analyzation process, including the geometrical and material parameters, is equal to the case studies in Section 3.1 and Section 3.2. The maximum stresses arise at the outer flank of the welding line, at the brace-to-chord and brace-to-brace connections (front side in Figure 13). Due to the axial compression without any imperfections, there is no failure by stability regarding the chord member. 

The results for the maximum compression force and the maximum deflection are printed in Figure 14 and Figure 15. Both graphs have a nearly linear and decreasing shape. The maximum range for the compression force is between 6.5 kN and 12.0 kN. However, by shortening the chord’s length from 3.0 m to 1.5 m, the applied compression force can be increased by a factor of 1.8 in the case of the 60° model. The delta of the deflection results is small. Generally, there is a reduction of 80% for the maximum compression force and 45% for the deflections in the case of the 60° inclination angle model.

### 3.3. FEM Analysis of the Expanded Geometry (Variable Geometry)

To estimate the influences of the expanded geometries, a second case study regarding expanded geometry is presented in this Section. The ratio of the expanded pipes’ lengths is changed, while the overall column length is fixed, as shown in Figure 16. Figure 16a–e show the model depending on the different length ratio. The chord length is set to 0.5 m to 2.5 m. The influence of variable chord and brace lengths on the maximum applied force in the elastic state is analyzed. 

The chord length varies in the range of 0.5 m to 2.5 m, staggered in 0.5 m intervals. The overall vertical column length is fixed to 3.0 m. Consequently, the length of the braces varies in the range of 0.5 m to 2.5 m equally to both members in vertical projection. This analysis is a theoretical study to figure out the behavior of the node in different geometrical situations compared with realistic geometries (Section 3.2). The ratio of the length between the chord and the brace is partly unrealistic for structures on site. The inclination angles are set to 30°, 45°, and 60°. Figure 16 shows the general geometry for the different length ratios in case of the 60° model.

The analyzation process, mesh, and load case are equal to the models in Section 3.1 and Section 3.2. Figure 17 visualizes the von-Mises stress distribution as an example for the 30° model including a chord length of 1.5 m. The maximum stresses arise at the welding line, between both branches. The forming of the stress peaks depends on the inclination angle.

Figure 18 shows the results for the maximum compression force for each brace, depending on the inclination angle and chord length. For all three model types, there is a quadratically increasing trend. The gap between results for the 30° and 45° model is smaller than for the 60° graph. The orientation of the trend is in contrast to the trend orientation of the results in Section 3.2. However, there is the same conclusion that if the member’s length increases, the resistance of the joint will decrease.

Figure 19 shows the results for the maximum deflections depending on the inclination angle and chord length. Mainly, there is a decreasing trend. The results of the 30° and 45° models are nearly superimposable, with an exception for the 0.5-m chord-length model. The 60° model has the largest deflections in comparison with the two other inclination angle models.

## 4. Discussion

The results presented in this paper show the relevance of the studies regarding symmetrical full-overlapped steel joints with on top connections. The Eurocodes [1] and Design Codes [2] disregard these geometries. The model presented in this paper is not comparable to the common systems in the standard. In the case of Y-joints or K-joints, there is an influence of the inclination angle on the maximum force. However, due to the intermediate plate and the different geometry, it is not possible to compare the maximum affected forces between both systems. In contrast with the conclusion of Saini [11], the welding line must be included to the numerical system, even if there are only axial forces. The highest stresses arise at the area around the welding line. 

Azari-Dodaran et al. [7] compared uniplanar and multiplanar steel joints. This paper presents uniplanar joints. By taking into account the results of the multiplanar joint of Heinemann et al. [3] and Azari-Dodaran et al. [7], the uniplanar models have a difference in resistance of −12% to +54%. 

The results regarding the influence of chord length on limit force are in contrast with the results of Espinosa et al. [8]. A large influence on the limit force occurred for a large ratio of the chord length. There was a maximum reduction of resistance of 80% in case of the expanded chord and fixed brace length. If the length of one member type is enlarged, the maximum affected force to the joint will decrease. If the overall length of the column is fixed, there will be a larger reduction of the maximum affected force compared with systems with unfixed geometries. Different failure modes of stability (buckling) arise for a small length ratio. This is characterized by a large pipe length and results in higher stresses or smaller applicable force to the joint. The cross-section diameters of the braces are smaller than the diameter of the chord. However, smaller cross-sections will have a smaller resistance to buckling effects, if there is a larger length. 

The influence of the inclination angle on the limit forces was investigated by comparing the results of this paper to the results of circular hollow-section joints with asymmetrical shape of Heinemann et al. [4]. The symmetrical joints have a difference in resistance of −13% to +25%, depending on the inclination angle.

Jankovic [8] presented a method to design hollow-section joints by calculating the interaction forces between the pipes. The connection forces are affected to the second pipe. The method is not applicable for the on-top-connection joint, which is presented in this paper. The welding line was neglected in Jankovic’s analyzation; however, the welding line represents the contact between the pipes. The stress distribution at the welding line is complex in the full-overlapped model. There is no possibility to calculate this distribution easily. The second aspect is about the overlapped geometry. A combination force of two pipes affects a third pipe regarding the model in this paper. This is not adoptable by Jankovic’s method. 

Mia et al. [9] conducted numerical case studies about XT-joints. No different inclination angles were tested. The stress distribution between the brace and the chord is comparable to the stress distribution in this paper. Besides this, the deformation behavior is comparable to the full-overlapped steel joints for the compression load case. No comparison regarding the force application of an overlapped joint can be done.

Rezadoost et al. [10] carried out studies about X-joints. It was found that the diameter ratio of the chord and brace has got a large influence on the resistance in the compression load case. A similar conclusion is drawn for the full-overlapped analysis by adopting the geometrical properties. By varying the inclination angle, the contact area between the braces and the top plate is enlarged. This variation has got an influence on the stress distribution at the top plate.

## 5. Conclusions

This paper presents numerical studies regarding full-overlapped columns made of circular hollow-section profiles with symmetric inclination angles. Small-scale models and different expanded models were analyzed. The following four conclusions are drawn: 

Firstly, the inclination angle has a large influence on the maximal force that can be applied to the structure. Besides this, the inclination angle has a large influence on the dimension of the maximal deflections. By increasing the inclination angle, the maximum force is increased by 85%, combined with an increase of 27% regarding the deflection. If the designing engineer chooses a full-overlapped column with on-top connection, the structure will be economically optimized by implementing an obtuse angle between the brace and horizontal level. This result is valid for structures with intermediate plates and unequal diameters for the brace and chord. 

Secondly, if there is a large chord length combined with a fixed brace length, the maximum load and global deformations will decrease. Depending on the inclination angle, the following influences occur by varying the chord length: 30° model—max. force −14%, deflections −12%; 45° model—max. force −25%, deflections −30%; 60° model—max. force −80%, deflections −45%. By comparing the results among the inclination angle models, the maximum resistance was observed to vary between 81% and 257%. The deflections rose to between 242% and 294% in maximum. In case of choosing a full-overlapped CHS column with on-top connection and fixed brace length, the most economical structure is archived by taking a short chord. In this case, the 1.5-m chord length modes enable the highest applicable force. 

Thirdly, if one pipe is enlarged, the resistance will decrease due to different failure modes. If the overall length is fixed and the pipe length becomes larger, the maximum affected force to the joint will be smaller than for the fixed-brace-length models. This reduction of the applied force occurs because of the smaller cross-section of the brace compared to the chord. The bending effect is larger in the case of a smaller cross-section diameter. This bending effect in case of large members with a small cross-section is explained in the Eurocode [1]. However, to archive the highest applicable force to a joint with full-overlapped top connection, the length of the pipe will be chosen as short and the ratio of the brace-to-chord length will tend to 1. This designing conclusion is valid for the three inclination angles, 30°, 45°, and 60°. 

Fourthly, regarding the enlarged models combined with a fixed overall length, the steep 60° model generated the highest compression force compared with the 30° and 45° systems. The 45° model is affected by the smallest force for this specific case. The areas including the highest stresses in the welding line are equal for both extracted systems. To choose an economical model, the designing engineer should pick an obtuse angle for this special kind of column.

The stress distributions are not comparable to current analyses in the literature (Section 4). Standard defined joints, such as the K- or Y- joints [8,9,10], are different in geometry. Due to the side connection, a different distribution of stresses arises in comparison to joints with intermediate plates and top connection. In comparison with multiplanar models [3,7], differences of up to 54% occur. The geometry including an intermediate plate reduces the resistance compared with joints with full-overlap (overlap of three members) [27]. In this case, the force is transferred directly into the chord and is not distributed in the intermediate plate. 

## Figures and Tables

**Figure 1 materials-15-03333-f001:**
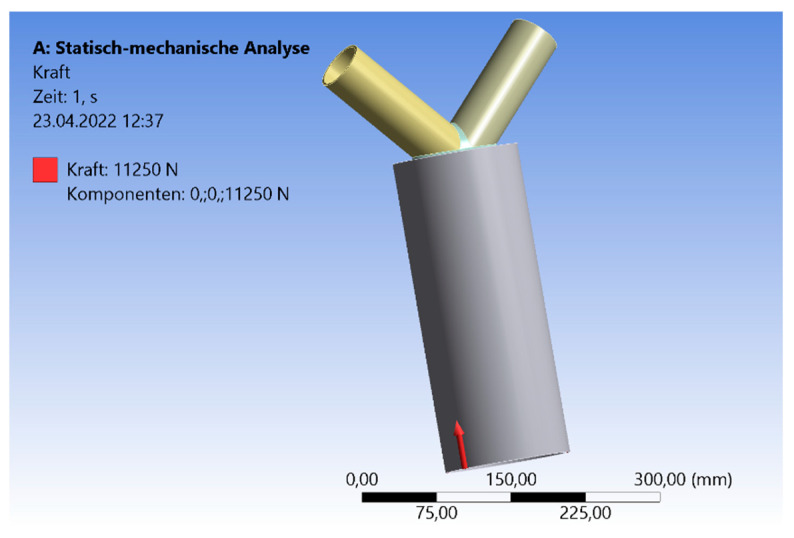
System and Load case CHS.

**Figure 2 materials-15-03333-f002:**
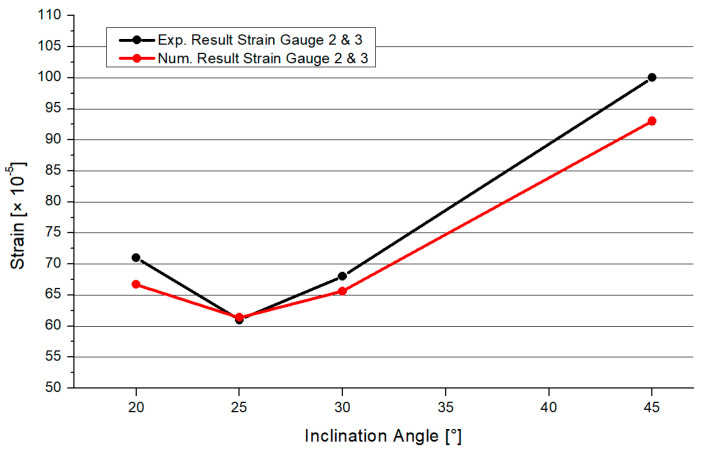
Validation Experimental–Numerical results [27].

**Figure 3 materials-15-03333-f003:**
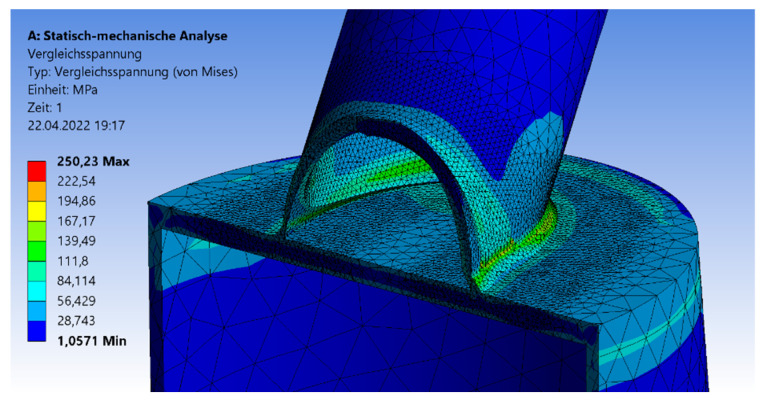
Von-Mises stress detailed over cross-section.

**Figure 4 materials-15-03333-f004:**
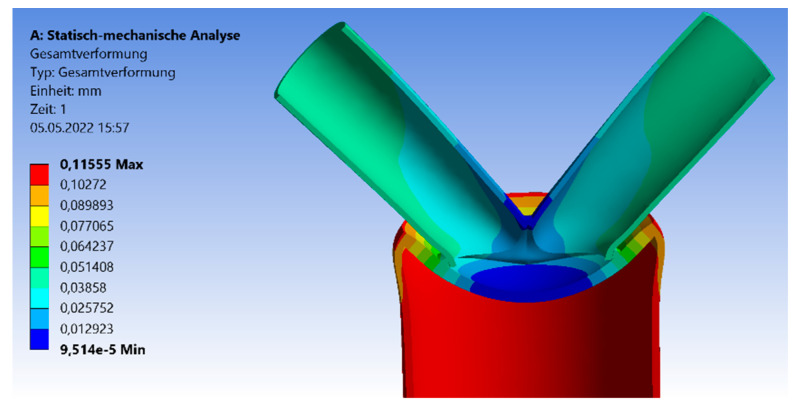
Deflection symmetrical 45° model—Section Cut.

**Figure 5 materials-15-03333-f005:**
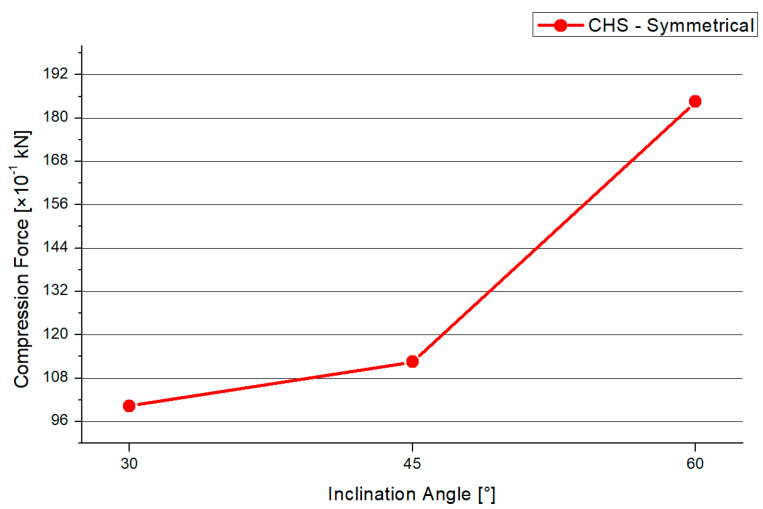
Maximum Compression Force CHS Symmetrical.

**Figure 6 materials-15-03333-f006:**
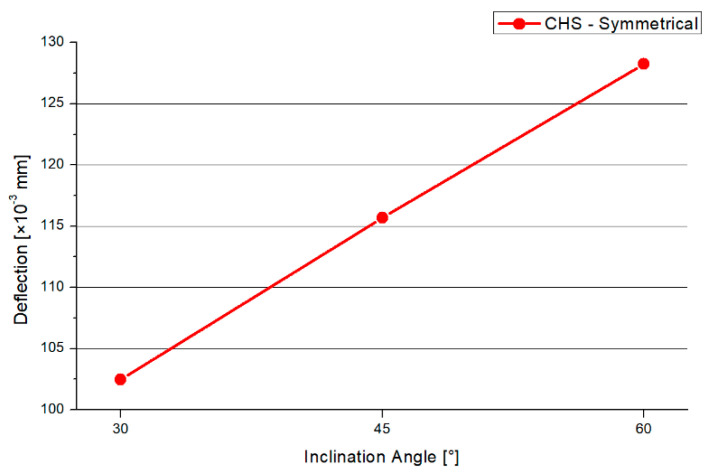
Deflection CHS Symmetrical.

**Figure 7 materials-15-03333-f007:**
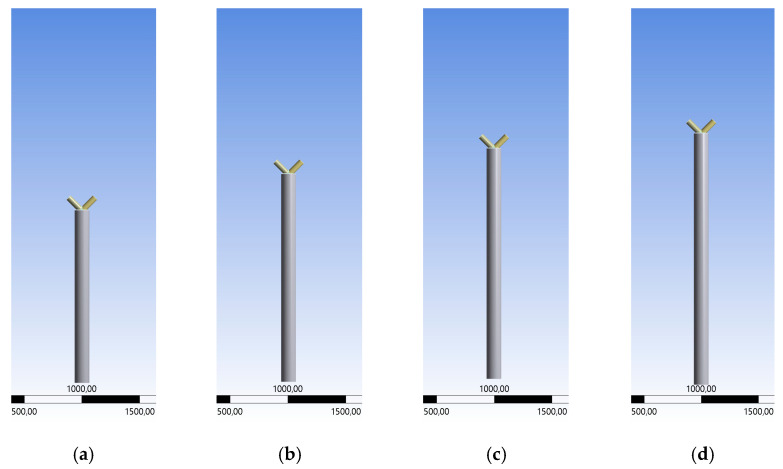
Variable Chord lengths, 1.5–3.0 m.

**Figure 8 materials-15-03333-f008:**
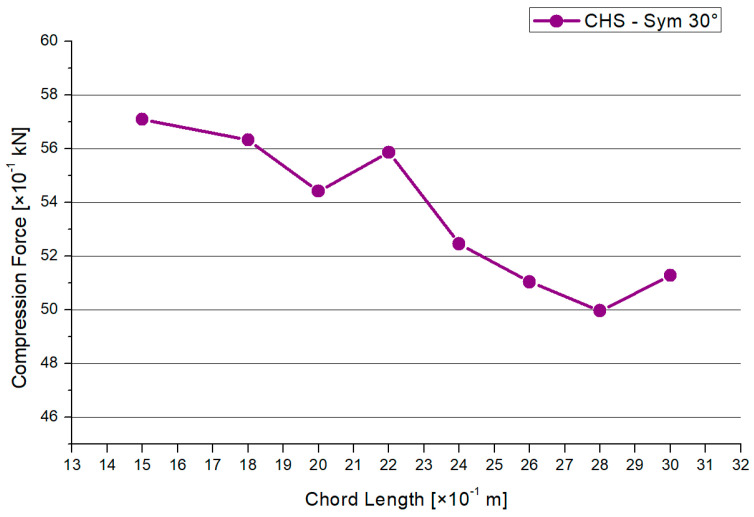
Results—Maximum Compression Force.

**Figure 9 materials-15-03333-f009:**
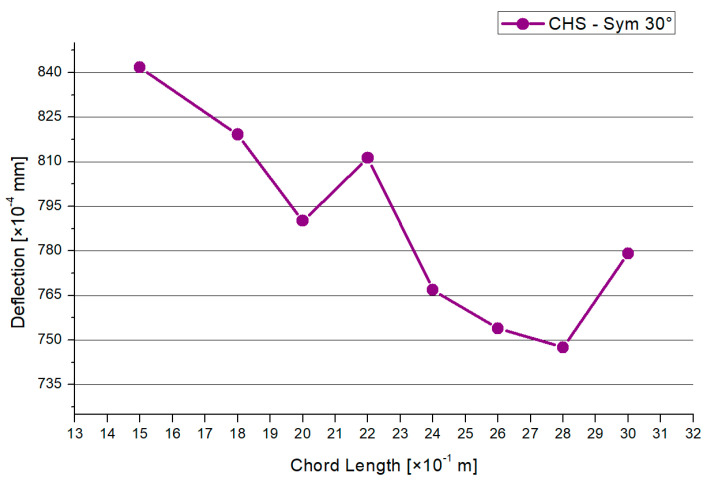
Results—Maximum Deflection.

**Figure 10 materials-15-03333-f010:**
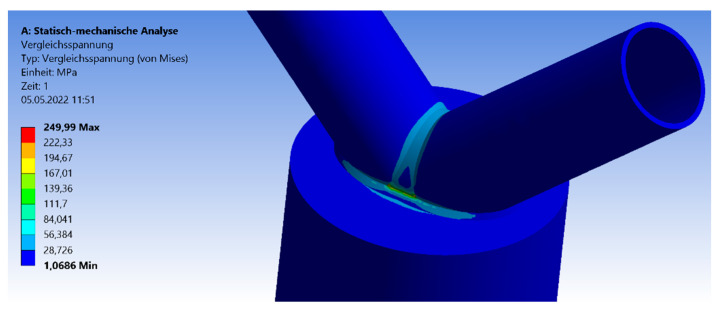
Stresses—symmetrical 45° model.

**Figure 11 materials-15-03333-f011:**
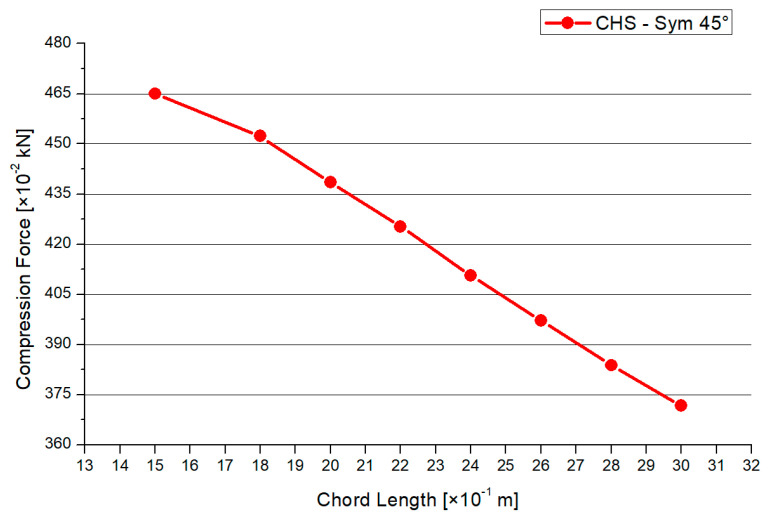
Results—Maximum Compression Force.

**Figure 12 materials-15-03333-f012:**
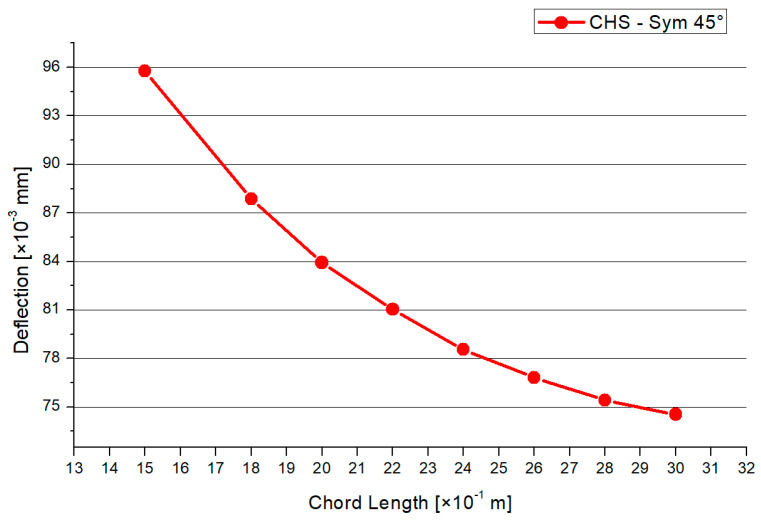
Results—Maximum Deflection.

**Figure 13 materials-15-03333-f013:**
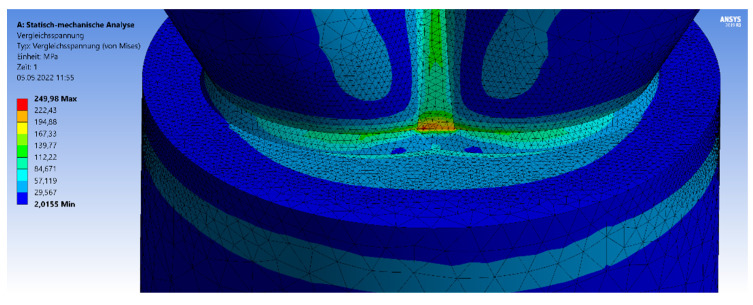
Stresses 60° Model.

**Figure 14 materials-15-03333-f014:**
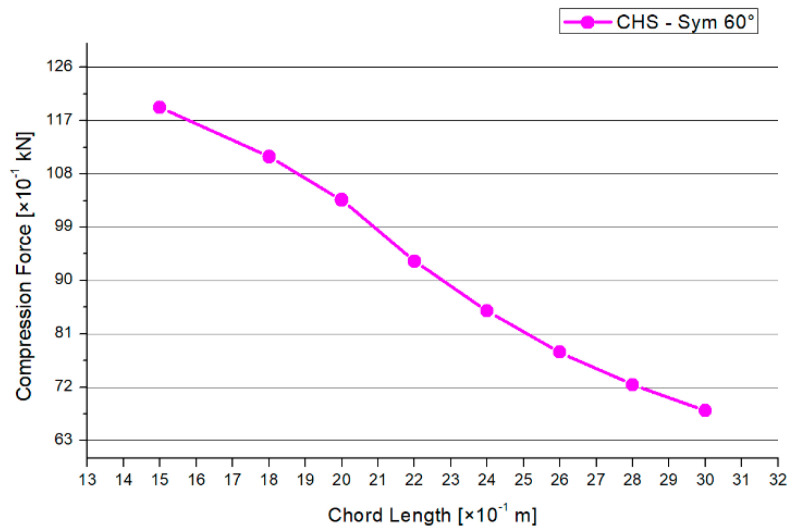
Results—Maximum Compression Force.

**Figure 15 materials-15-03333-f015:**
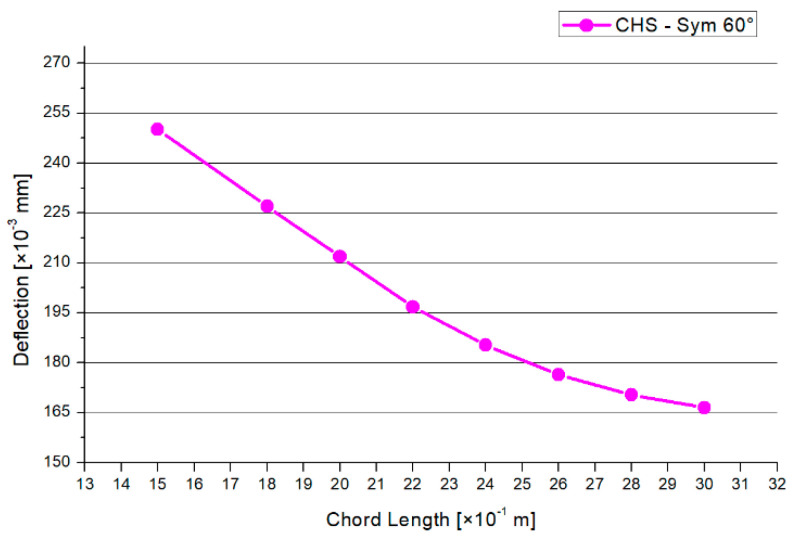
Results—Maximum Deflection.

**Figure 16 materials-15-03333-f016:**
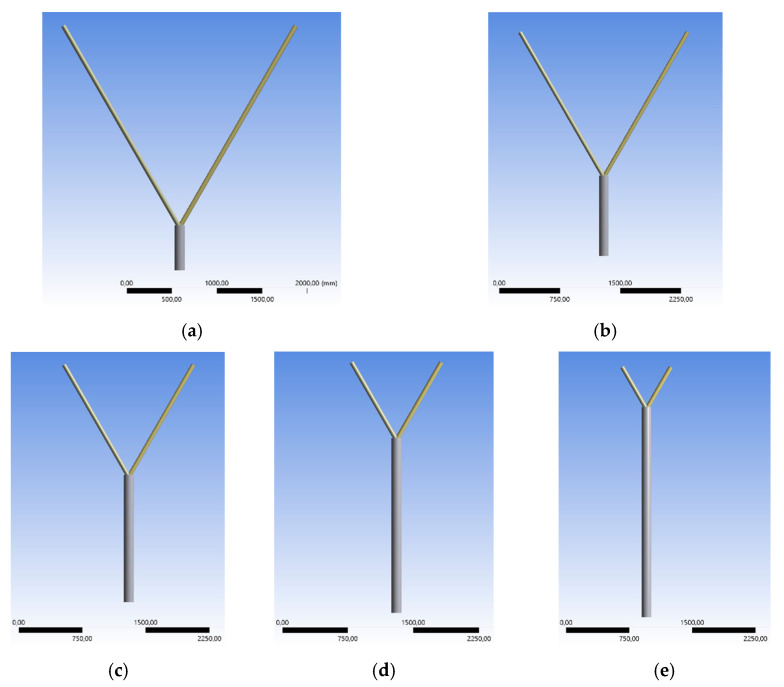
Variable Geometry with Chord Length 0.5–2.5 m.

**Figure 17 materials-15-03333-f017:**
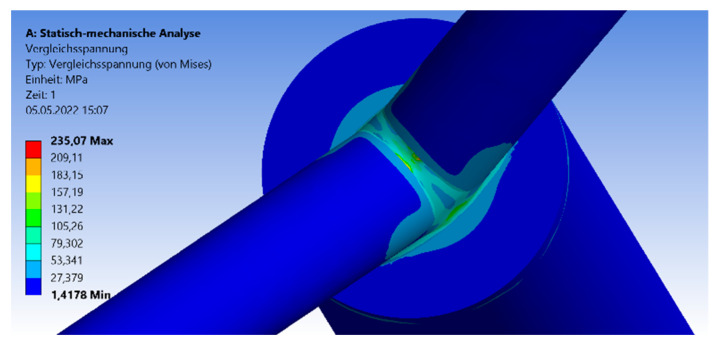
Von-Mises Stress Distribution—1.5-m chord length.

**Figure 18 materials-15-03333-f018:**
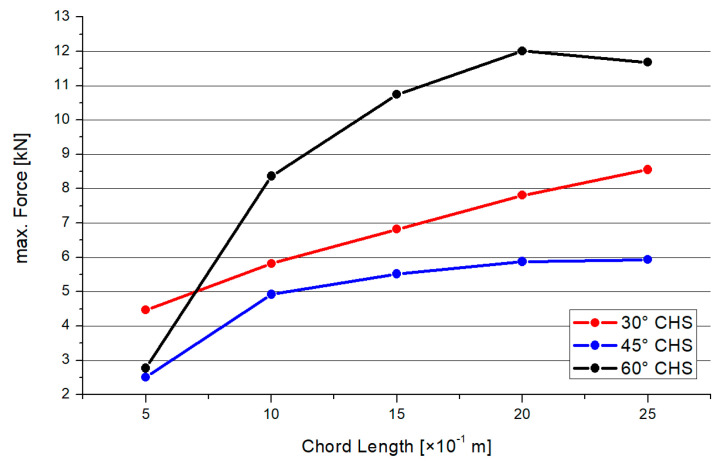
Results—Maximum Compression Force.

**Figure 19 materials-15-03333-f019:**
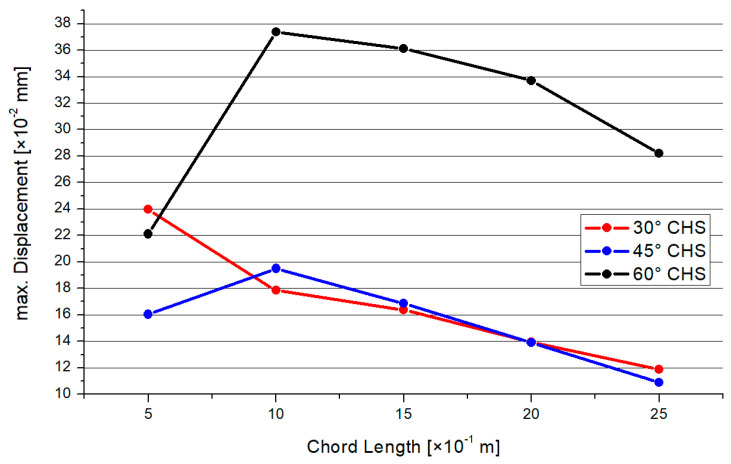
Results—Maximum Deflection.

## Data Availability

Not applicable.

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
