# Peer review of "Numerical Case Studies about Two-Dimensional CHS Joints with Symmetrical Full-Overlapped Top-Connection"

_materials, 2022, doi:10.3390/ma15093333_

Round 1
Reviewer 1 Report
This manuscript deals with the case study of joins of pipes. The effects of the lengths and the angle of the connected pipes were investigated. They found that the inclination angle of 60°ã€€can bear highest load and obtuse angle is economical. The information might be of interest to the readers of Materials. It looks there is no serious problem in methodology. I have only minor comments listed below:
- Line 126, what does “simply supported” mean? Is it mean the displacement of the edge of braces were fixed but rotations were allowed?
- Line 129 to 130, how did the load apply to the model? It maybe uniformly applied to the edge nodes of the chord so that the sum of the load becomes target value, but it should be clearly explained to ensure reproducibility.
- Line 140, please explain the details of material’s model including material parameters. Is the analysis elastic or elastoplastic? If it is elastoplastic, which model was used?
- Line 147, figure 2 do not visualize the refined elements of welding line. Please add the figure which visualize the refined elements or remove the sentence.
- Line 173 (the caption of Figure 2), the reference seems to be wrong. I guess ref. 24 is correct reference.
- Figure 3 and hereafter, please add color bar which shows the magnitude of the von-mises stress.
- Line 197, “ultimate limit state of the joint” has not been defined before this line. Elastic limit state is better for reader’s understanding as line 234.
- Figure 7, if the authors would like to show the difference of the chord length, it is better to use the same magnification scale to the Figs. 7 (a-h).
- Line 231, Figure 7 doesn’t show the deformation in case of 30° model. Please add the figure or remove the sentence.
-
It would attract much attention of the readers If the authors could deduce more general guideline.
Author Response
Dear Reviewer 1, first of all we want kindly thank you for the effort in analyzing the paper and the feedback. We really appreciate it. All comments, which were listed in the review are adopted in the new version of the paper.
Point 1: Line 126, what does “simply supported” mean? Is it mean the displacement of the edge of braces were fixed but rotations were allowed?
Response 1: Yes, you are right. In opposite to fixed braces, where bending is transferred, simple supported members have got a blocked displacement. We wanted to adopt the models as close to the comparable experimental tests, where the structure is based on a platform.
Point 2: Line 129 to 130, how did the load apply to the model? It maybe uniformly applied to the edge nodes of the chord so that the sum of the load becomes target value, but it should be clearly explained to ensure reproducibility.
Response 2: The load affects the cross section area of the chord. This is more realistic, than the force application to the edge node in view of a compression force machine. So, the force is transformed by ANSYS from N to N/mm² automatically. In the results we only estimated the total force as a sum of the above described forces.
Point 3: Line 140, please explain the details of material’s model including material parameters. Is the analysis elastic or elastoplastic? If it is elastoplastic, which model was used?
Response 3: The analysis is executed in the elastic state. The main properties of the material are added to the article.
Point 4: Line 147, figure 2 do not visualize the refined elements of welding line. Please add the figure which visualizes the refined elements or remove the sentence.
Response 4: You are right, the notation was wrong. The right figure is “Figure 3”, where the mesh/refined mesh and the stress distribution can be found.
Point 5: Line 173, (the caption of Figure 2), the reference seems to be wrong. I guess ref. 24 is correct.
Response 5: You are completely right, we are sorry. Due to the addition of some references, the new notation is 27.
Point 6: Figure 3 and hereafter, please add color bar which shows the magnitude of the von-misses stress.
Response 6: The color bars are added for all relevant figures (Figure 3, 4, 10, 13, 17).
Point 7: Line 197, “ultimate limit state of the joint” has not been defined before this line. Elastic limit state is better for reader’s understanding as line 234.
Response 7: The term “elastic limit state” is added to this position.
Point 8: Figure 7, if the authors would like to show the difference of the chord length, it is better to use the same magnification scale to the Figs. 7 (a-h)
Response 8: The pictures in Figure 7 are changed due to better clearness. All pictures have got the same magnification scale to estimate the chord length in a better way. Only the model (without deflection) is visualized due to the small scale.
Point 9: Line 231, Figure 7 doesn’t show the deformation in case of 30° model. Please add the figure or remove the sentence.
Response 9: You are right, this sentence confused the reader. It is removed.
Point 10: It would attract much attention of the readers if the authors could deduce more general guideline.
Response 10: This paper is part of a research study, where several versions of full overlapped joints are investigated under different geometrical properties. In case of this paper, the subject is concentrated on the inclination angle and the influence of the chord length on the resistance of this special joint. The conclusion of this paper regarding the choice of the inclination angle and chord length should help the designing engineer to optimize the structure economically.
Thank you very much and kind regards,
The authors

Reviewer 2 Report
The article presents current engineering issues. The paper was prepared in a thoughtful and correct manner. The content of the publication does not raise any objections. The paper is appropriately divided into several main sections, which are summarized with adequate research results and conclusions. The literature presented in the paper is sufficient. In my opinion, the content of the work does not require modification and the article itself can be published after minot corrections:
- the introduction has been written in a correct manner, but the novelty of this paper in relation to other thematically similar research papers should be demonstrated.
2. the introduction should cite advanced numerical simulations using FEM, due to the importance of this technique in solving engineering problems, citing works such as (10.1016/j.compositesb.2020.107931, 10.1002/nme.6757, 10.1016/j.compositesb.2013.10.080) - figure 7 is presented in an unreadable manner. Please improve the way the essence of the figure is presented.
- in figures 10, 13, 17 the lower scale should not overlap the figure as it is unreadable.
- conclusions should better present a quantitative and qualitative assessment of the research findings.
Author Response
Dear Reviewer 2, first of all we want kindly thank you for the effort in analyzing the paper and the feedback. We really appreciate it. All comments, which were listed in the review are adopted in the new version of the paper.
Point 1: The introduction has been written in a correct manner, but the novelty of this paper in relation to other thematically similar research papers should be demonstrated.
Response 1: Generally, it is hard to find literature about these special kinds of steel joints. Mostly, variations of standard-defined joints are analysed. The most similar steel joints are Y-or K-joints, which are already mentioned in the article. The geometrical and loading properties are not comparable to the joint type described in the article. This joint type is commonly used in practise. But there is no designing recommendation for the designing engineer. This article should help the designing engineer to choose the most economical strucure under the aspect of the inclination angle and it underlines the novelty of the topic. We added some more details about this item. The article is part of a research study, where standard-undefined steel joints are analysed.
Point 2: The introduction should cite advanced numerical simulations using FEM, due to the importance of this technique in solving engineering problems, citing works such as (10.1016/j.compositesb.2020.107931, 10.1002/nme.6757, 10.1016/j.compositesb.2013.10.080)
Response 2: Literature regarding FEM analyses is included. Thank you for the examples concerning FEM studies. Due to the short time to improve the paper, only few articles can be considered.
Point 3: Figure 7 is presented in an unreadable manner. Please improve the way the essence of the figure is presented.
Response 3: The visual style of Figure 7 is improved. The magnification scale is equal to all pictures in this figure to compare the differences in length of the chord. Figure 7 is an example for the general model. The scale is too small presenting deflections (comment of other reviewers), so only the model is shown in this figure.
Point 4: In Figures 10, 13, 17 the lower scale should not overlap the figure as it is unreadable.
Response 4: The scale is removed in some figures. Because of the visualisation of details in this figure (for example, the mesh) it is not possible to include the scale outside of the figures. But the scale can be seen as an example in Figure 17.
Point 5: Conclusions should better present a quantitative and qualitative assessment of research findings.
Response 5: Please find the comparison of the results of the article in the discussion chapter. Generally, it is hard to compare the conclusions to the literature. Due to the different geometry or load cases, stress distributions differ. A paragraph is added to the conclusion chapter, where the different stress distributions are explained. Beside this, the conclusion about the bending effect of large-chord-length models are compared to the Eurocode.
Thank you very much and kind regards,
The authors

Reviewer 3 Report
Reviewer's notes:
This paper deals with the numerical case studies of two-dimensional circular hollow-section joints in order to assess the resistance typical steel joints. The subject of investigation is generally interesting. However, some corrections should be made in order to better present conducted investigation:
- In Chapter 2 Authors state that that gap is between 1-2 mm. Figures 1 and 3 that there is larger space between pipes. Please explain with more details.
- In Figure 1 load cases should be presented either by ANSYS side legend or by text marks
- Generally, Figures should follow explanation text explaining figures not vice versa. This should be checked for all Figures.
- In Chapter 2, Line 147-148, it is stated that refinement is shown on Figure 2. Figure 2. Shows validation results. This should be corrected by adding another figure.
- In Chapter 2, Line 170, strain gauge position is mentions and results for strain gauge 2 and 3 are presented on Figure 2. Strain gauge placement and numbering should be explained and shown in order to clearly compare results.
- Reference 21 shown in description of Figure 2 does not include mentioned results. Check this reference.
- Figure 3 and all Figures with results of FEA (stress or displacement/deflection) should have Legend with stress values in order to clearly present stress intensity.
- Check spelling should be done to correct spelling errors like: und in line 74.
Author Response
Dear Reviewer 3, first of all we want kindly thank you for the effort in analyzing the paper and the feedback. We really appreciate it. All comments, which were listed in the review are adopted in the new version of the paper.
Point 1: In chapter 2 Authors state that that gap is between 1-2 mm. Figures 1 and 3 that there is larger space between pipes. Please explain with more details.
Response 1: In this context the gap is defined between the top plate to brace or brace to brace connection. On the construction site, commonly there is a gap between the pipes, before they are welded. Due to the welding line a fixed connection between the pipes is created. To adopt this process in the numerical model, the connection between the pipes is set as frictionless. The connection between the pipes and the welding is set as fixed. So, we make sure, there is no contact between the pipes numerically.
So, this setting can not be seen in the figures. Due to the sectional view in Figure 3 it looks like, as if there is a large gap. But actually there is no gap in the numerical model. You can see it in Figure 4 (deflection), that the geometry of the braces are directly on the top plate. However, it is only a numerical adoption. We added an explanation about this numerical adoption.
Point 2: In Figure 1 load cases should be presented either by ANSYS side legend or by test marks.
Response 2: In Figure 1, the side legend of the axial compression force is added as an example of one inclination angle.
Point 3: Generally, Figures should follow explanation text explaining figures not vice versa. This should be checked for all Figures.
Response 3: The position of the figures are new arranged, following the explanation text, under the aspect of the layout. So, sometimes figures have to be combined to complete the layout of a page.
Point 4: In chapter 2, Line 147-148, it is stated that refinement is shown on Figure2. Figure 2 shows validation results. This should be corrected by adding another figure.
Response 4: You are right, the notation of the figure was wrong. Instead of figure 2 it must be figure 3. There the mesh/refined mesh and the distribution of the stresses can be found.
Point 5: In chapter 2, Line 170, strain gauge position is mentioned and results for strain gauge 2 and 3 are presented on Figure 2. Strain gauge placement and numbering should be explained and shown in order to clearly compare results.
Response 5: The position of the strain gauges is described in the article, now. This figure is an example of one position. Strain gauges 2 and 3 are superimposable due to the geometry. Both strain gauges are fixed on the top position of the vertical welding line on both brace surfaces. One strain gauge is fixed on brace 1 and the second strain gauge is fixed on brace 2. Strain gauge 1 is fixed at another position and excluded in this article.
Point 6: Reference 21 shown in description of Figure 2 does not include mentioned results. Check this reference.
Response 6: You are right, the reference is corrected now. The reference must be ref. 24 instead of ref. 21. I am sorry.
Point 7: Figure 3 and all figures with results of FEA (stress or displacement/deflection) should have legend with stress values in order to clearly present stress intensity.
Response 7: The legends are added to the figures including the stress or distribution of deflection. The stress is iterated up to the limit of 250 MPa, which is the elastic limit of the chosen material.
Point 8: Check spelling should be done to correct spelling errors like: und in line 74
Response 8: The spelling is checked and the word is corrected.
Thank you very much and kind regards,
The authors
